# A Two-Dimensional Adaptive Target Detection Algorithm in the Compressive Domain

**DOI:** 10.3390/s19030567

**Published:** 2019-01-29

**Authors:** Wenhuan Cao, Shucai Huang

**Affiliations:** Air and Missile Defense College, Air Force Engineering University, Xi’an 710051, China; hsc67118@126.com

**Keywords:** compressive domain, compressive sensing, compressive subtraction, two-dimensional measurement model, adaptive threshold method

## Abstract

By applying compressive sensing to infrared imaging systems, the sampling and transmitting time can be remarkably reduced. Therefore, in order to meet the real-time requirements of infrared small target detection tasks in the remote sensing field, many approaches based on compressive sensing have been proposed. However, these approaches need to reconstruct the image from the compressive domain before detecting targets, which is inefficient due to the complex recovery algorithms. To overcome this drawback, in this paper, we propose a two-dimensional adaptive threshold algorithm based on compressive sensing for infrared small target detection. Instead of processing the reconstructed image, our algorithm focuses on directly detecting the target in the compressive domain, which reduces both the time and memory requirements for image recovery. First, we directly subtract the spatial background image in the compressive domain of the original image sampled by the two-dimensional measurement model. Then, we use the properties of the Gram matrix to decode the subtracted image for further processing. Finally, we detect the targets by employing the advanced adaptive threshold method to the decoded image. Experiments show that our algorithm can achieve an average 100% detection rate, with a false alarm rate lower than 0.4%, and the computational time is within 0.3 s, on average.

## 1. Introduction

Compressive sensing (CS)-based target detection is a hot topic in remote sensing. Two different methods are available to resolve this problem: reconstruction-based detection and handling the image in the compressive domain without reconstruction. Aiming at a one-dimensional measurement model, previous studies [1,2,3] detected and classified targets based on reconstructing the image. Reconstruction algorithms for the two-dimensional (2D) measurement model are priorities in recent research [4,5,6,7,8,9]. Although these reconstruction-based detection algorithms achieved CS-based target detection, they do not only impact the detection results, but also require more memory space and operation time. Detection algorithms in the compressive domain have increasingly attracted attempts to enhance detection efficiency while avoiding the effects of reconstruction algorithms. Although the research outputs are still limited, Ma et al. [10] proposed a conventional energy detection-based non-reconstruction CS detection algorithm, and Xu et al. [11] and Li et al. [12] realized compressive detection by means of directly decoding targets’ spatial positions in the compressive domain. All these algorithms were designed for one-dimensional measurement models, which merely achieve compressive sampling of row vectors while the column vectors keep their original dimensions. Target detection based on 2D measurement models that achieve compressive sampling of both row and column vectors remains a challenge.

To achieve 2D measurement model-based target detection in the compressive domain, we propose a 2D compressive domain adaptive threshold (2D CDAT) algorithm that uses Gram matrix mapping of the compressive subtracted image from the compressive domain to the space domain without reconstructing the image, and detects targets via an advanced adaptive threshold method.

This paper is organized as follows: firstly, Section 2 illustrates the CS measurement models, especially the two-dimensional measurement model. Our 2D CDAT algorithm is then outlined in Section 3. Finally, Section 4 provides the experimental results.

## 2. Compressive Sensing—The Basics

The CS theory was proposed by Donoho in 2006 [13]. After being vigorously developed, CS theory became a hot research topic in the applied mathematics and signal processing fields. A combined compression and sampling method for sparse or compressible signals was the core concept of CS theory; in other words, the signal was compressed and sampled simultaneously at a rate below the Nyquist rate.

Traditional one-dimensional measurement models only compress and sample the column vectors of the signal, whereas the row vectors of the signal are not compressed. If we want to further compress the signal, we can use a 2D measurement model to compress the signal rows and columns simultaneously [14,15].

Supposed that a two-dimensional signal $X\in R^{N_{1}\times N_{2}}$, called *K* (*K < N*_1_, *K* < *N*_2_) ranks sparse, meaning ***X*** only has *K* nonzero elements. The compressive sampling signal $Y\in R^{M_{1}\times M_{2}}$ can be calculated through two measurement matrixes, $\Phi_{c}\in R^{M_{1}\times N_{1}}$ and $\Phi_{r}\in R^{M_{2}\times N_{2}}$, which were non-adaptive; they can simultaneously compress a signal’s row and column vectors separately. The compressive sampling ratios of rows and columns are *R_r_* = *M*_2_/*N*_2_ and *R_c_* = *M*_1_/*N*_1_, respectively, corresponding to the compressive sampling ratio of the 2D measurement model, which is $R_{CS2}=\left(M_{1}\times M_{2}\right)/\left(N_{1}\times N_{2}\right)$. Thereby, the 2D measurement equation can be expressed as:(1)$$Y=\Phi_{c}X\Phi_{r}^{T}$$
where every element in ***Y*** contains the complete information of ***X***, as shown in Figure 1. Compared with the one-dimensional measurement model, the 2D measurement model can compress the signal’s row and column vectors simultaneously, which more deeply compresses the image, and lowers the compressive sampling ratio. Additionally, the detector’s scale is further decreased, and the memory capacity requirement in the hardware implementation is reduced.

Natural signals are not usually sparse in the time domain, whereas if the original signal is sparse or compressible in the ***Ψ*** domain, Equation (1) can be written as:(2)$$Y=\Phi_{c}X\Phi_{r}^{T}=\Phi_{c}\Psi_{c}S\Phi_{r}^{T}\Psi_{r}^{T}=A_{c}SA_{r}^{T}$$
where ***A****_c_* and ***A****_r_* are the sensing matrix, and ***Ψ****_c_* and ***Ψ****_r_* are sparsity bases. There are several sparsity bases, including the discrete cosine transform (DCT) base, the fast Fourier transform (FFT) base, the discrete wavelet transform (DWT) base, the Curvelets base [16], the Gabor base [17], and redundant dictionaries [18].

CS is an under-sampling measurement, and there are infinite solutions satisfying Equation (2) because the dimensions of ***Y*** are much fewer than for ***X***. Therefore, dissolving the original signal ***X*** from the measurement data ***Y*** is a nondeterministic-polynomial (NP)-hard problem. To reconstruct the sparse signal precisely, Candes and Tao [19] proved that the sensing matrix ***A*** must satisfy the restricted isometry property (RIP). Giving a discretional value of *k =* 1,2,…,*K*, we defined ***A***’s restricted isometry constants *a_k_* to be the smallest quantity, such that ***A*** obeys:(3)$$\left(1-a_{k}\right){\left\|x\right\|}_{2}^{2}\leq {\left\|Ax\right\|}_{2}^{2}\leq \left(1+a_{k}\right){\left\|x\right\|}_{2}^{2}$$
where *x* is a *k*-order sparse constant, $0<a_{k}<1$, and ***A*** satisfies *k*-order RIP. We can design the measurement matrix to make ***A = ΦΨ*** meet the RIP requirement on the condition that ***Ψ*** is constant. There are three kinds of random measurement matrixes [20] that are frequently used: (1) Measurement matrix elements are independent, and they obey a certain distribution, including the Gauss Random matrix, Bernoulli Random matrix, and Sub-Gauss Random matrix. (2) The measurement matrix is composed of random rows in any orthogonal matrix, including the Part Fourier matrix, Part Hadamard matrix, and Noncorrelation Random matrix. (3) The measurement matrix is composed of one specific signal, including the Toeplitz matrix, Circulant matrix, Binary Sparse matrix, and Structurally Random matrix.

## 3. The 2D CDAT Target Detection Algorithm

The main idea of 2D CDAT first involved subtracting the compressive image with a background, and then mapping the compressive-subtracted image to the spatial domain by using the property of the Gram matrix, then finally detecting targets via the adaptive threshold partition method.

### 3.1. Background Subtraction Method

Space-based geosynchronous Earth orbit early warning satellites run in geosynchronous orbit with an orbital altitude ranging from 36,000 km to 40,000 km. For an optical imaging system with a focal length of 1 m and a pixel size of 30 μm, the corresponding ground distinguishability is 1 km [21]. The weak flame of a ballistic missile is tens of meters wide and hundreds of meters long, and the Earth’s surface is covered with clouds within an altitude of 20 km. Hence, targets usually occupy one or several pixels in the image, and clouds mainly occupy the rest. Supposing a detector images at a speed of two frames per meter; each pixel in the image represents an area of 1 km^2^ when detecting a ballistic missile at a speed of 4–8 km/s, so its speed in the image plane is 2–4 pixels per frame. Suppose that the maximum drift speed of clouds is 50 m/s, so that its speed in the image plane is 0.025 pixels per frame; the clouds can be considered as static or slowly changing in the background. Therefore, the frame subtraction method, which sees the previous frame image as a background, then obtains the subtraction image of the current frame and background, and can reserve targets and remove most of the background points. Suppose the current frame image is ***X****_t_*, the previous frame image is ***X****_t_*_–1_, and the subtraction image is ***X****_d_*; their relationship is expressed as:(4)$$X_{d}=X_{t}-X_{t-1}$$

***X****_d_* reserves the foreground targets and a small amount of mutational background points, so it is sparser and has a higher signal-to-noise ratio (SNR) than ***X****_t_*.

A regular mask compressive sampling pattern, using the same measurement matrixes ***Φ***_c_ and ***Φ***_r_ to achieve the 2D measurement $Y_{d}=\Phi_{c}X_{t}\Phi_{r}^{T}$, can be used for sequential images in compressive sensing. The following can be obtained from background subtraction in the compressive domain:(5)$$Y_{d}=(Y_{t}-Y_{t-1})=\Phi_{c}(X_{t}-X_{t-1})\Phi_{r}^{T}$$

If the grayscale of the mutational background points was extremely high, these mutational background points were likely to be mistaken for targets in the subtraction image. We had to update the background using the current frame image to reduce the influence of the mutational background points and implement the adaptive background update. The updated background is as follows:(6)$$Y_{t-1}^{*}=\alpha Y_{t}+(1-\alpha )Y_{t-1}$$
where *α* is the learning rate of the background-adaptive update. Hence, the subtraction image in the compressive domain can be further rewritten as:(7)$$Y_{d}^{*}=Y_{t}-Y_{t-1}^{*}=(1-\alpha )(Y_{t}-Y_{t-1}^{})$$

### 3.2. Mapping from the Compressive Domain to the Spatial Domain

Analyzing the normalized measurement matrix $\Phi \in R^{M\times N}$, given that the Gram matrix ***G*** is $G=\Phi^{T}\Phi$, $G\in R^{N\times N}$, the diagonal elements of ***G*** are all 1, and the rest of the elements are small values of approximately 0, meaning ***G*** ≈ ***I****_N_*, where ***I*** is a unit matrix, and the higher the dimensions of the ***G*** matrix, the smaller the values of the off-diagonal elements. Taking the Gauss Random Matrix as an example, to statistically analyze ***G***’s off-diagonal elements, 100 tests were completed with different dimensions matrixes. The Probability Density Function (PDF) of ***G***’s off-diagonal elements is shown in Figure 2. With the expansion of ***G*** matrix’s dimensions, the probability of off-diagonal elements near 0 increased, and the ***G*** matrix was increasingly close to the unit matrix. Therefore, the Gram matrix can be used to extract target spatial information in compressive images.

The size of the Gram matrix is proportional to the compressive sampling ratio. Usually, the images used in the experiment were 512 × 512 or 256 × 256 pixels, the compressive sampling ratio of the rows and columns was at least 0.5, and the image size was compressed to at least one- quarter of the original. The corresponding measurement matrix was 256 × 512 or 128 × 256 pixels. Therefore, the size of the Gram matrix was 256 × 256 or 128 × 128, respectively. In subsequent experiments, only a 128 × 128 Gram matrix was used, so Figure 2 only shows a 256 × 256 Gram matrix.

The following operations are performed on $Y_{d}^{*}$: (8)$$\begin{array}{cc} Z_{t} & =\Phi_{c}^{T}Y_{d}^{*}\Phi_{r}\\ & =(1-\alpha )\Phi_{c}^{T}\Phi_{c}(X_{t}-X_{t-1})\Phi_{r}^{T}\Phi_{r}\\ & =(1-\alpha )G_{c}(X_{t}-X_{t-1})G_{r} \end{array}$$

The SNR of the subtraction image was high because there were only foreground targets and a few mutational background points in the subtraction image. The subtraction values were larger at the target positions, while being close to 0 in other positions. The Gram matrix was close to the unit matrix; as a result, the Gram matrix could approximately recover the subtraction image. ***Z****_t_* did not change the characteristics of the subtraction image (***X****_t_* – ***X****_t_*_–1_) after operation, where the targets were still outstanding, and the original zero-value points were approximately 0 after the operation. Hence, the compressive subtracted image could be mapped in the spatial domain using Equation (8), where the subtraction of the original image was recovered and the targets’ spatial information was mapped out. Then, the targets’ positions can be detected from ***Z****_t_* using an appropriate threshold.

### 3.3. Adaptive Threshold Partition

In target detection, it is usually necessary to set a threshold to distinguish the target from the background to identify the background as being below the threshold and the target as being higher than the threshold. The gray values of the targets and background points were different in every frame image because the targets and the background points were changing at every moment in the sequential images. Therefore, the fixed threshold partition was bound to cause a higher probability of false alarm. The adaptive threshold is set from the point of probability and statistics by calculating the mean and standard deviation of the gray level of the image. This threshold is not fixed—it varies with the change in the means and standard deviations of different images, and so is referred to as an adaptive threshold.

The mean and variance of the recovery compressive-subtracted image of the current image were calculated: ***Z****_t_* = {*z_ij_* | *i =* 1,2,…, *N*_1,_
*j* = 1,2,…,*N*_2_} in the spatial domain *E*(***Z****_t_*) = *μ*_0_, $D\left(Z_{t}\right)=\sigma_{0}^{2}$. According to the 3*σ* property of the Chebyshev inequality, the probability of ***Z****_t_* is:(9)$$P\left(\mu_{0}-3\sigma_{0}<Z_{t}\leq \mu_{0}+3\sigma_{0}\right)\approx 0.9974$$

Points are more likely to be background points if *z_ij_*ϵ [ *μ*_0_ – 3*σ*_0_, *μ*_0_ + 3*σ*_0_] and to be targets if *z_ij_*> *μ*_0_ + 3*σ*_0_. Hence, *Th*_0_ = *μ*_0_ + 3*σ*_0_ can be used as an initial partition threshold of the foreground and the background, as shown in Equation (10), with points larger than *Th*_0_ being reserved as the foreground, and points less than *Th*_0_ being set to 0 as the background:(10)$$z_{ij}=\{\begin{array}{l} z_{ij}\\ 0 \end{array}\begin{array}{cc} & \begin{array}{c} z_{ij}>Th_{0}\\ z_{ij}\leq Th_{0} \end{array} \end{array}$$

The threshold needs to be further improve because some background points with high grayscale difference are mistaken as foreground points after image subtraction, which can lead to a higher probability of false alarm. The mean and variance of the foreground reserved by the initial threshold method were *μ* and *σ*^2^, respectively. Most of the foreground points were mutational background points for small targets, but the targets were more prominent than the background points. Therefore, the threshold can be improved using *Th* = *μ* + *σ*^2^ to filter more background points. Similarly, according to the 3*σ* property of the Chebyshev inequality, the range of *k* was [−3,3]. As shown in Equation (11), the *z_ij_* points in the recovery spatial subtraction image were deemed to be abnormal points when they were larger than *Th* and they were set to 1 as potential targets, whereas the points less than *Th* were set to 0 as the background:(11)$$z_{ij}=\{\begin{array}{c} 1\\ 0 \end{array}\begin{array}{cc} & \begin{array}{c} z_{ij}>Th\\ z_{ij}\leq Th \end{array} \end{array}$$

Setting up different *k* values will lead to different probabilities of detection and false alarm. The larger the *k* value, the larger the detection threshold, which will lead to a lower probability of detection and false alarm. In engineering applications, the sizes of the *k* values can be set according to the probability of false alarm requirements. There was no requirement for the false alarm rate in the following experiments. In order to verify the universality of the algorithm, we chose k=0 as a compromise.

## 4. Simulation and Analysis

The simulation data were derived from a real sky background infrared image collected by a medium-wave infrared thermal imager, and the imager’s parameters are shown in Table 1. The acquisition time was the morning of 5 May 2016, and the weather was cloudy. After the 2976th frame, there were two targets of about 16 and 11 pixels in the infrared image.

The 2980th frame was selected as the current image, and a 256 × 256 pixel size image, including targets, was captured as experimental data, as shown in Figure 3.

We subtracted the 2980th from the 2979th frame, shown in Figure 4a, in which the background points were filtered effectively after background subtraction, and the SNR of the image improved. Then, the effect of the method used to map from the compressive domain to the spatial domain was analyzed, and different types of measurement matrixes were chosen to verify the universality of the algorithm. First, Gauss, Part Hadamard, Bernoulli, Circulant, and Toeplitz were used as measurement matrixes for compressing and sampling the sequential images, with the measurement matrixes $\Phi_{c}\in R^{M_{1}\times N_{1}}$, $\Phi_{r}\in R^{M_{2}\times N_{2}}$, where *M*_1_ = *M*_2_ = 128, *N*_1_ = *N*_2_ = 256, and the compressive sampling ratio *R_CS_*_2_ = 0.25. After compressive sampling, the size of the images was changed to 128 × 128 pixels. Then, the subtraction images in the compressive domain were mapped to the spatial domain, and the sizes of the images were recovered to 256 × 256. As shown in Figure 4b–f, the Gram matrix could effectively recover the subtraction image. The recovery subtraction image in which target points were prominent and the SNR was high could be used to further detect the targets.

Five indexes, including SNR, the Signal Clutter Ratio (SCR), the Background Suppression Factor (BSF), the Receiver Operating Characteristics (ROC), and the Area Under the Curve (AUC), were used to evaluate the mapping effect of the subtraction image. The definitions of these indexes are as follows:(12)$$\mathrm{SNR}=\left|\frac{\mu_{t}-\mu_{b}}{\sigma_{b}}\right|$$
where *μ*_t_ is the mean of the target gray level and *μ*_b_ is the mean of the background gray level, and *σ*_b_ is the standard deviation of the background gray level. SNR mainly reflects the correlation between the target gray level and the background gray level. The larger the SNR, the smaller the correlation between the target and the background, and the less the target is disturbed by the background:(13)$$\mathrm{SCR}=\left|\frac{\mu_{t}-\mu_{b}}{\mu_{t}+\mu_{b}}\right|$$
where *μ*_t_ is the mean of the target gray level and *μ*_b_ is the mean of the background gray level. SCR mainly reflects the difference between the target gray level and the background gray level. The bigger the SCR, the bigger the gray difference between the target and the background:(14)$$\mathrm{BSF}=\frac{\sigma_{\mathrm{in}}}{\sigma_{\mathrm{out}}}$$
where *σ*_in_ and *σ*_out_ are the standard deviations of background gray level before and after filtering, respectively. The larger the BSF, the stronger the suppression of the background after filtering.

For the filtered residual image, the ROC curve of the detection algorithm can be drawn by changing the detection threshold *T* and traversing the probability of false alarm *P*_f_ to obtain the corresponding probability of detection *P*_d_, taking *P*_f_ as the horizontal axis and *P*_d_ as the longitudinal axis. *P*_d_ and *P*_f_ are defined as:(15)$$\begin{array}{cc} P_{d}=\frac{N_{t}}{S_{t}}, & P_{f}=\frac{N_{b}}{S_{b}} \end{array}$$
where *N*_t_ represents the number of pixels that the algorithm detects as the correct target, *S*_t_ represents the number of real pixels of the target, *N*_b_ represents the number of pixels where the algorithm detects the wrong target, and *S*_b_ represents the number of real pixels of the background. Under the same *P*_f_, if the *P*_d_ of the algorithm is higher than that of the others, it means the algorithm has better performance.

The area AUC under the ROC curve can be divided into several trapezoids. Let the point on the ROC curve be set as (*x_i_, y_i_*)(*i* = 1,…,*n*), where *n* is the total number of points on the ROC curve. AUC can be expressed as:(16)$$\mathrm{AUC}=\frac{1}{2}\sum_{i=1}^{n}\left(x_{i}-x_{i-1}\right)\left(y_{i}+y_{i-1}\right)$$

The larger the AUC value, the better the performance of the algorithm and the better the detection performance. The average ROC curve of the 100 experiments was compared due to the randomness of the measurement matrix. Figure 5 depicts the average ROC curve of the 100 tests.

The recovery subtraction image was compared with the original 2980th frame image, and the evaluation indexes are shown in Table 2. The table shows that the images mapped to the spatial domain through the measurement matrix were far higher in SNR and SCR than in the original 2980th frame image. The SNR values in the Part Hadamard measurement and the Circulant measurement were higher, which indicates that the correlation between the target and the background is smaller, and that the targets are less disturbed by the background. The values of SCR and BSF in the Part Hadamard measurement and Toeplitz measurement were higher, which indicates that the difference between the gray level of the targets and the background is larger, and that the background is strongly suppressed after filtering. The AUC value of the Part Hadamard measurement and Toeplitz measurement was higher, which shows that the two measurements are better in general, and that they produce better detection performance.

The detection effect of the adaptive threshold partition was further analyzed. Figure 6a–e are images of the initial threshold partition, in which many false alarm points exist. The accuracy of detection decreased because the accumulation of false alarm points was mistaken for targets in small target detection. 

Figure 6f–j are the adaptive threshold partition images when k=0, in which the false alarm points were filtered effectively and did not accumulate. Targets in the adaptive threshold partition images were still outstanding. This shows that the adaptive threshold partition method could effectively reduce the probability of false alarms by guaranteeing the detection rate.

Compared the proposed 2D CDAT algorithm to the traditional detection algorithm using reconstruction, the 2D field was established using the Part Hadamard Random matrix model and the compressive subtracted image was reconstructed using the 2D iterative adaptive approach (2D IAA) algorithm [4] and the 2D Smoothed L0 (2D SL0) algorithm [5]. The sparse base was the DCT base. The parameters were set to LL = 10 and Ite = 500 in the 2D IAA algorithm, and to L = 3 in the 2D SL0 algorithm after repeated experiments to achieve better results. k=0 was set in the 2D CDAT algorithm. Figure 7a,b shows the reconstruction of the compressive subtracted image, and Figure 7d,e show the target detection results produced by the 2D SL0 algorithm and the 2D IAA algorithm, respectively. 

Figure 7c,f compare the 2D CDAT algorithm, the 2D IAA algorithm, and the 2D SL0 algorithm. Each algorithm was tested 100 times to reduce the random error, and the average ROC curve was drawn as shown in Figure 8.

Table 3 shows the operation time, AUC, SNR, SCR, and BSF of each algorithm. The operation time, AUC, SNR, and SCR of the 2D CDAT algorithm were superior to the 2D IAA and 2D SL0 algorithms because the 2D SL0 algorithm and the 2D IAA algorithm needed many iterative optimizations during the reconstruction process. As the 2D CDAT algorithm needed no reconstruction to directly map the compressive subtracted image to the spatial domain, it was more efficient. AUC showed that the 2D CDAT algorithm was better overall. SNR and SCR indicated that targets in the subtraction image were more prominent and obvious after being mapped to the spatial domain by the 2D CDAT algorithm and that the difference between the targets and the background increased.

However, the BSF value for the 2D CDAT algorithm was lower than that of the 2D SL0 and the 2D IAA algorithms, which shows that the background was restrained weakly. This was due to the high sparsity of the compressive subtracted image. The original subtraction image can be recovered efficiently through the reconstruction of the compressive subtracted image using the 2D SL0 and 2D IAA algorithms, whereas 2D CDAT algorithm was mapped to the compressive subtracted image using the property of the Gram matrix, and could not completely recover the original subtraction image.

The focus was to separate the target from the background for the target detection problem. We did not need to excessively pursue the recovery degree of the image. From the detection results, the adaptive threshold detection method used by the 2D CDAT algorithm could effectively filter the false alarm points by guaranteeing the detection rate compared with the 2D SL0 and 2D IAA algorithms.

## 5. Conclusions

The main contributions and conclusions our work are as follows:(1)2D CDAT was proposed to detect a target without reconstruction. Aiming at a 2D measurement model, the signal was mapped from the compressive domain to the spatial domain by directly using the characteristics of the Gram matrix, and the advanced adaptive threshold partition method was used to detect the target.(2)Using real infrared images as the experimental data, the compressive subtracted images were mapped to the spatial domain using different measurement matrixes. Experiments showed that the decoded subtraction images produced higher SNR and SCR than the original images. The comparison of the mapping effect of the observation matrixes indicated that the subtraction images mapped by the Part Hadamard matrix achieved higher SNR, SCR, BSF, and AUC values, and better detection performance.(3)Comparing the 2D CDAT algorithm with the traditional reconfiguration detection algorithms, three algorithms—2D CDAT, 2D IAA, and 2D SL0—were used to detect the target. The results showed that the 2D CDAT algorithm was superior to the other two algorithms in terms of operation time, AUC, SNR, and SCR, and that it could achieve more efficient target detection.(4)The proposed 2D CDAT algorithm can be used in remote sensing field to achieve a more efficient target detection. In the next step, we consider to apply the 2D CDAT algorithm to the target tracking issues.

## Figures and Tables

**Figure 1 sensors-19-00567-f001:**
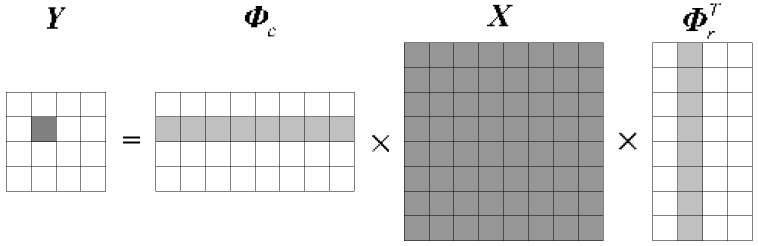
Two-dimensional (2D) measurement model graph: ***Y*** is the compressive sampling signal, ***Φ****_c_* and ***Φ****_r_* are measurement matrixes, ***X*** is the two-dimensional original signal.

**Figure 2 sensors-19-00567-f002:**
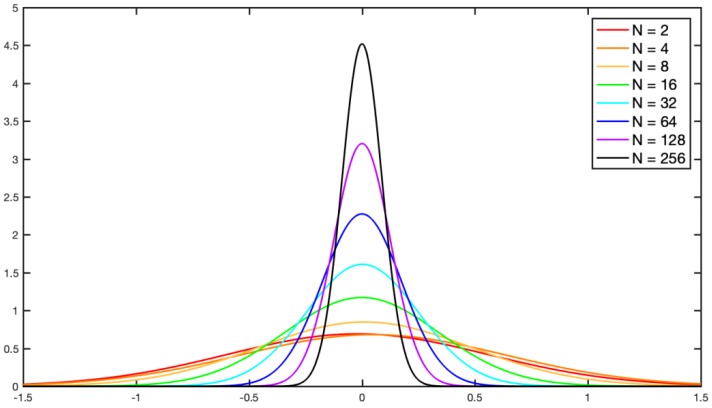
Probability density function of the Gram Matrix’s off-diagonal elements: the dimension N of the Gram matrix is from 2 to 256.

**Figure 3 sensors-19-00567-f003:**
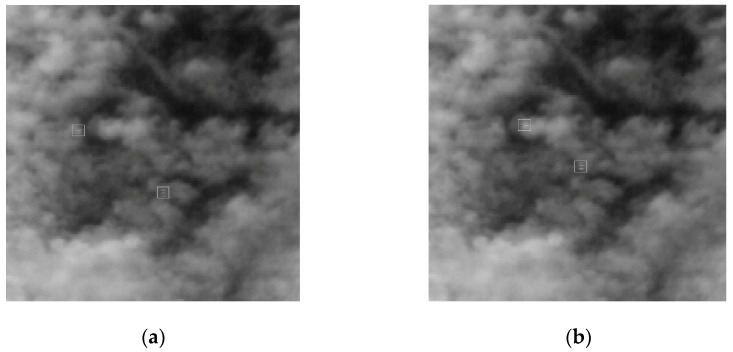
Experimental images: The backgrounds are clouds that are hiding the targets. The targets are marked with a white square frame, representing 16 and 11 pixels. (**a**) 2979th frame image and (**b**) 2980th frame image.

**Figure 4 sensors-19-00567-f004:**
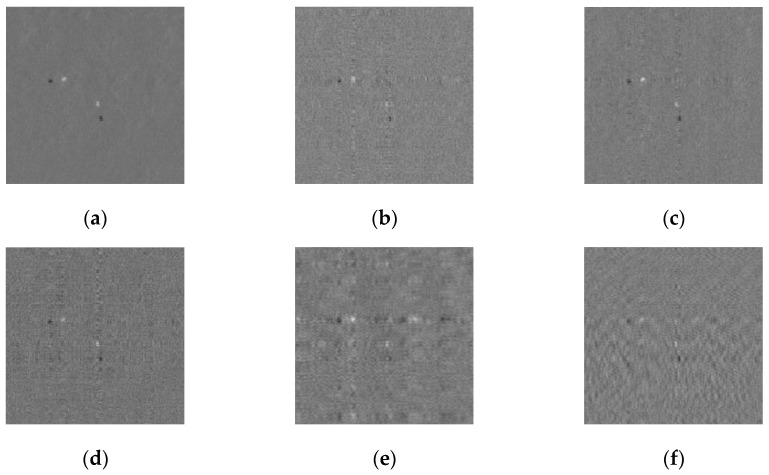
Subtraction image mapping by different measurement matrixes: The targets in the space domain mapping images are more prominent compared with the original image. (**a**) Real spatial subtraction image, (**b**) Space domain mapping image by Gauss, (**c**) Space domain mapping image by Part Hadamard, (**d**) Space domain mapping image by Bernouli, (**e**) Space domain mapping image by Circulant, (**f**) Space domain mapping image by Toeplitz.

**Figure 5 sensors-19-00567-f005:**
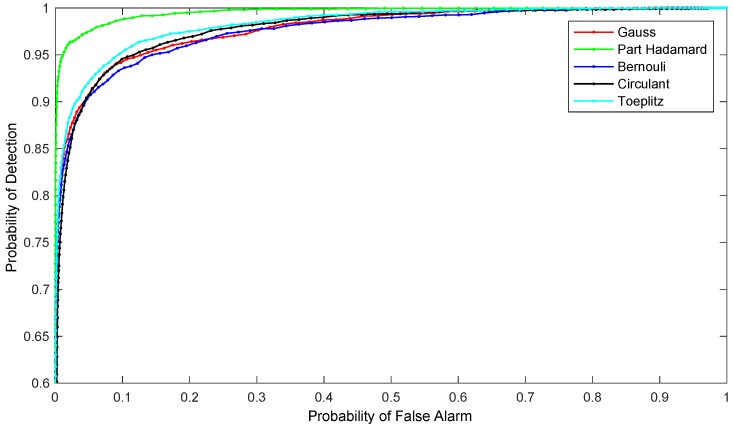
The average ROC curve of subtraction image mapping by different measurement matrixes: the measurement matrixes are Gauss matrix, Part Hadamard matrix, Bernouli matrix, Circulant matrix and Toeplitz matrix.

**Figure 6 sensors-19-00567-f006:**
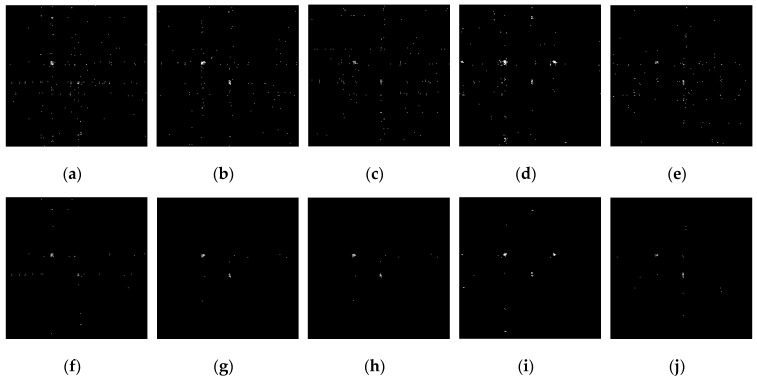
The threshold partition image by the initial threshold and the adaptive threshold of different measurement matrixes: (**a**) Initial threshold partition image mapping by Gauss, (**b**) Initial threshold partition image mapping by Part Hadamard, (**c**) Initial threshold partition image mapping by Bernouli, (**d**) Initial threshold partition image mapping by Circulant, (**e**) Initial threshold partition image mapping by Toeplitz, (**f**) Adaptive threshold partition image mapping by Gauss, (**g**) Adaptive threshold partition image mapping by Part Hadamard, (**h**) Adaptive threshold partition image mapping by Bernouli, (**i**) Adaptive threshold partition image mapping by Circulant, (**j**) Adaptive threshold partition image mapping by Toeplitz.

**Figure 7 sensors-19-00567-f007:**
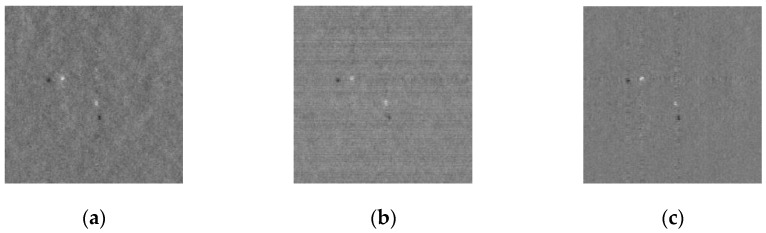
Comparison of algorithms of the recovered spatial domain image, and detection results: (**a**) Compressive subtracted image reconstructed by 2D SL0, (**b**) Compressive subtracted image reconstructed by 2D IAA, (**c**) Compressive subtracted image mapping by Part Hadamard, (**d**) Detection result of 2D SL0, (**e**) Detection result of 2D IAA, (**f**) Detection result of 2D CDAT.

**Figure 8 sensors-19-00567-f008:**
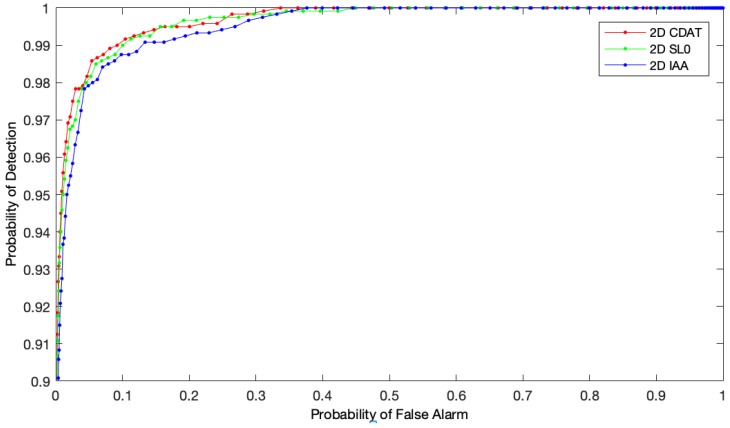
The average ROC curves of the compressive subtracted image recovered by different algorithms: the algorithms are 2D CDAT, 2D SL0 and 2D IAA.

**Table 1 sensors-19-00567-t001:** Medium-wave infrared thermal imager parameters.

Parameter	Details	Parameter	Details
Wide field of view	13.75 × 11°	Charge-coupled device (CCD)	480 × 640
Narrow field of view	2.29 × 1.83°	Frame frequency	50 Hz
Pixel size	30 × 30 μm	Power	28 W
Focal length	40–240 mm	Noise equivalent temperature difference (NETD)	≤35 mK
Working waveband	3–5 μm	Minimum resolvable temperature difference (MRTD)	≤0.3 K

**Table 2 sensors-19-00567-t002:** Evaluation index of the subtraction image.

Image	Signal-to-Noise Ratio (SNR)	Signal Clutter Ratio (SCR)	Background Suppression Factor (BSF)	Area Under the Curve (AUC)
2980th frame image	1.038	0.169	—	—
Gauss	4.888	1.016	46.18	0.9779
Part Hadamard	8.307	1.064	103.25	0.9966
Bernouli	4.415	1.021	45.62	0.9757
Circulant	7.027	1.015	40.58	0.9781
Toeplitz	4.483	1.035	57.13	0.9827

**Table 3 sensors-19-00567-t003:** Comparison of the algorithm performance of different indexes.

Algorithms	Operation Time (s)	AUC	SNR	SCR	BSF
2D CDAT	0.26	0.9966	8.307	1.064	103.25
2D SL0	8.24	0.9962	7.641	1.009	1.54 × 10^3^
2D IAA	13.93	0.9950	6.619	1.007	780.66

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
