# Peer review of "A Two-Dimensional Adaptive Target Detection Algorithm in the Compressive Domain"

_sensors, 2019, doi:10.3390/s19030567_

Round 1
Reviewer 1 Report
I will not recommend this paper to be published in the current form. There are too many grammar errors in it and make it difficult to follow. For example, the second sentence in the abstract “Aiming at two-dimensional measurement model, ….” needs to be rewritten. Next sentence should be “Simulation experiment demonstrated …” I will review it again if they can do better proofreading and resubmit. Some suggestions for the authors: 1. “compressive domain” might be a better description than “non-reconstruction”. 2. I am not sure why the authors felt like they need to have a review of 1D measurement model since this paper is about compressive imaging?Author Response
Response to Reviewer 1 Comments
Dear Reviewer:
Thank you for your comments concerning our manuscript entitled “Two-Dimensional Non-Reconstruction Compressed Sensing Adaptive Target Detection Algorithm” (Manuscript ID: sensors-411437). Those comments are all valuable and very helpful for revising and improving our paper, as well as the important guiding significance to our researches. We have studied comments carefully and have made correction which we hope meet with approval. Revised portion are marked in red in the paper. The main corrections in the paper and the responds to the reviewer’s comments are as flowing:
Point 1: I will not recommend this paper to be published in the current form. There are too many grammar errors in it and make it difficult to follow. For example, the second sentence in the abstract “Aiming at two-dimensional measurement model, ….” needs to be rewritten. Next sentence should be “Simulation experiment demonstrated …” I will review it again if they can do better proofreading and resubmit.
Response 1: We regret there were problems with the English. We have rewritten the abstract and the paper has been carefully revised by a professional language editing service to improve the grammar and readability.
Point 2: “compressive domain” might be a better description than “non-reconstruction”.
Response 2: We have made correction according to the Reviewer’s comments. We have deleted “non-reconstruction” and replaced by “compressive domain”. The title of the article was revised to read “Two-Dimensional Adaptive Target Detection Algorithm in the Compressive Domain”. And the name of our algorithm was revised to “two-dimensional compressive domain adaptive threshold (2D CDAT) algorithm”.
Point 3: I am not sure why the authors felt like they need to have a review of 1D measurement model since this paper is about compressive imaging?
Response 3: We have re-written this part according to the Reviewer’s suggestion. In Section 2, we have deleted the review of 1D measurement model, and put the describe of sparsity bases measurement matrixes at the end of Section 2.
Other changes:
1. The numbering style of section changed from “I, II,......” to “1,2,......”.
2. The Tables and Figures were added with more detailed caption to make them clear.
3. The Author Contributions and the Conflicts of Interest were added at the end of the manuscript.
We tried our best to improve the manuscript and made some changes in the manuscript. These changes will not influence the content and framework of the paper. And here we did not list the changes but marked in red in revised paper. We appreciate for Reviewer’ warm work earnestly, and hope that the correction will meet with approval.
Once again, thank you very much for your comments and suggestions.

Reviewer 2 Report
The authors present a 2D non-reconstruction compressed sensing algorithm to detect sparse targets: fast-moving ballistic missiles in cloudy sky. In the proposed algorithm, the authors first produced a background subtracted image, then the background subtracted image was mapped to spatial domain with Gram matrix, finally adaptive thresholding was utilized for decision making in target detection.
The overall organization of the paper is clear, however within individual sections the writing is unclear, both due to sentence-level organizational structure as well as grammatical errors, misspellings, typos, and undefined acronyms (NETD, MRTD, IAA, SL0, DCT), which need to be fixed to make the paper easier to read. This is especially important in Section III (which is particularly unclear) where the authors describe their new approach, the 2DNRAT algorithm.
In addition, more detailed captions can be added to relevant figures and tables in order to make them easier to understand.
There are also some mathematical errors that need to be corrected. For example, on lines 86 and 88, the dimensions of the X, Phi_c and Phi_r matrices are given. But given these dimensions, the equation on line 92 doesn’t make sense. Perhaps the transpose should be removed in line 92. Furthermore, on line 113, the authors say, “the max drift speed of clouds was 50 km/s”. This seems far too fast for clouds.
In Section IIIB, the authors explain that larger Gram matrices provide better reconstructions of the difference image. However, they do not discuss that larger Gram matrices would seem to correspond to worse compression ratios. Could the authors explain why they stop at N = 256 instead of an even larger value to make use of even narrower PDF in Figure 2? Presumably, this has to do with compression ratio too. A discussion of this tradeoff should be included. Furthermore, in section IV, no mention of compression ratios are given. The compression ratio is a critical parameter when evaluating different compression matrices, as the authors do in Figs. 4-8.
In section IIIC, the authors set mu_0+3*sigma_0 as initial partition threshold and change to mu+k*sigma^2 (assuming the authors meant mu+k*sigma) to filter more background points. However, how the authors decide on what value of parameter k to ultimately use is not discussed. What value of k is used in Figs. 6 and 7? I think how to decide the value for k is important as the algorithm is called “adaptive target detection algorithm”. Is k the parameter that is swept in constructing the ROC curves in Fig. 5 and 8, or is it some other parameter? This should be made clear in the text.
In section IV, five indices, SNR, SCR, BSF, ROC, and AUC are used to evaluate different methods. It would be beneficial to include the mathematical equations for how they are defined.
At the end of the manuscript, the author contributions are not filled out.
I would recommend some more citations too:
On line 25, where the authors cite 2D reconstruction algorithms, I think it would be good to cite some of the recent microscopy reconstruction work:
Y. Rivenson, Y. Wu, H. Wang, Y. Zhang, A. Feizi, and A. Ozcan, “Sparsity-based multi-height phase recovery in holographic microscopy,” Sci. Rep. 6(1), 37862 (2016).
C. Fournier, F. Jolivet, L. Denis, N. Verrier, E. Thiebaut, C. Allier, and T. Fournel, “Pixel super-resolution in digital holography by regularized reconstruction,” Appl. Opt. 56(1), 69–77 (2017).
Zhen Xiong, Jeffrey E. Melzer, Jacob Garan, and Euan McLeod, “Optimized sensing of sparse and small targets using lens-free holographic microscopy,” Optics Express, 26 (20), 25676-25692 (2018).
On lines 73-79, references for these different matrices should be given. If they are all covered in one or two review articles, or a textbook, that would suffice.
On line 230, references for the IAA and SL0 algorithm should be given.
Author Response
Response to Reviewer 2 Comments
Dear Reviewer:
Thank you for your comments concerning our manuscript entitled “Two-Dimensional Non-Reconstruction Compressed Sensing Adaptive Target Detection Algorithm” (Manuscript ID: sensors-411437). Those comments are all valuable and very helpful for revising and improving our paper, as well as the important guiding significance to our researches. We have studied comments carefully and have made correction which we hope meet with approval. Revised portion are marked in red in the paper. The main corrections in the paper and the responds to the reviewer’s comments are as flowing:
Point 1: The overall organization of the paper is clear, however within individual sections the writing is unclear, both due to sentence-level organizational structure as well as grammatical errors, misspellings, typos, and undefined acronyms (NETD, MRTD, IAA, SL0, DCT), which need to be fixed to make the paper easier to read. This is especially important in Section III (which is particularly unclear) where the authors describe their new approach, the 2DNRAT algorithm.
Response 1: We regret there were problems with the English. The paper has been carefully revised by a professional language editing service to improve the grammar and readability. The acronyms NETD and MRTD were defined in Table 1, IAA and SL0 were defined in the Section 4, and DCT was defined in the Section 2.
Point 2: In addition, more detailed captions can be added to relevant figures and tables in order to make them easier to understand.
Response 2: We have made correction according to the Reviewer’s comments. The tables and figures were added with more detailed caption to make them clear.
Point 3: There are also some mathematical errors that need to be corrected. For example, on lines 86 and 88, the dimensions of the X, Phi_c and Phi_r matrices are given. But given these dimensions, the equation on line 92 doesn’t make sense. Perhaps the transpose should be removed in line 92.
Response 3: We are very sorry for our incorrect writing, it should be and we revised it. After revised the the dimensions of , the equation(1) on line 92 was right and the transpose needed be kept. And we have reviewed all matrices and equation to avoid this problem.
Point 4: Furthermore, on line 113, the authors say, “the max drift speed of clouds was 50 km/s”. This seems far too fast for clouds.
Response 4: We are very sorry for our incorrect writing, the max drift speed of clouds was 50 m/s, and we have revised in the Section 3.1.
Point 5: In Section IIIB, the authors explain that larger Gram matrices provide better reconstructions of the difference image. However, they do not discuss that larger Gram matrices would seem to correspond to worse compression ratios. Could the authors explain why they stop at N = 256 instead of an even larger value to make use of even narrower PDF in Figure 2? Presumably, this has to do with compression ratio too. A discussion of this tradeoff should be included.
Response 5: Considering the Reviewer’s suggestion, we have given a supplementary statement in Section 3.2. “The size of Gram matrix is proportional to the compressive sampling ratio. Usually, the image used in the experiment is 512×512 or 256×256 pixels, the compressive sampling ratio of rows and columns is at least 0.5, and the image size is compressed to 1/4 of the original at least. The corresponding measurement matrix is 256×512 pixels or 128×256 pixels. Therefore, the size of Gram matrix is 256×256 or 128×128, respectively. In subsequent experiments, only Gram matrices of 128×128 size are used, so Figure 2 is only shown in Gram matrix of 256×256 size.” was added in the Section 3.2.
Point 6: Furthermore, in section IV, no mention of compression ratios are given. The compression ratio is a critical parameter when evaluating different compression matrices, as the authors do in Figs. 4-8.
Response 6: We are very sorry for our negligence of compression ratios. We have added “The compressive sampling ratios of rows and columns are and , respectively” in Section 2, and added “First, Gauss, Part Hadamard, Bernoulli, Circulant, and Toeplitz were used as measurement matrixes for compressing and sampling the sequential images, with the measurement matrixes ,, where , , and the compressive sampling ratio . After compressive sampling, the size of the images was changed to 128 × 128 pixels. Then, the subtraction images in the compressive domain were mapped to the spatial domain, and the sizes of the images were recovered to 256 × 256.” in Section 4.
Point 7: In section IIIC, the authors set mu_0+3*sigma_0 as initial partition threshold and change to mu+k*sigma^2 (assuming the authors meant mu+k*sigma) to filter more background points. However, how the authors decide on what value of parameter k to ultimately use is not discussed. What value of k is used in Figs. 6 and 7? I think how to decide the value for k is important as the algorithm is called “adaptive target detection algorithm”. Is k the parameter that is swept in constructing the ROC curves in Fig. 5 and 8, or is it some other parameter? This should be made clear in the text.
Response 7: It is really true as Reviewer suggested that how to decide the value for is important, we have re-written the Section 3.3 according to the Reviewer’s suggestion.
In target detection, it is usually necessary to set a threshold to distinguish the target from the background, to identify the background below the threshold and the target higher than the threshold. The gray values of targets and background points were different in every frame image because the targets and background points were changing at every moment in sequential images, therefore, fixed threshold partition was bound to cause higher probability of false alarm. While the adaptive threshold is set from the point of probability and statistics by calculating the mean and standard deviation of the gray level of the image. This threshold is not fixed. It varies with the change of the mean and standard deviation of different images, so it is called adaptive.
Adaptation is due to mean and standard deviation, not to the change of . In order to further enhance the validity of the threshold, the factor is added to further distinguish the target from the background. “Similarly, according to property of Chebyshev inequality, the range of is [-3,3]” was added in the Section 3.
“Setting up different values will lead to different probability of detection and false alarm. The larger the value, the larger the detection threshold, which will lead to lower probability of detection and false alarm. In engineering applications, the size of values can be set according to the requirements of probability of false alarm. There is no requirement for false alarm rate in the following experiments. In order to verify the universality of the algorithm, we choose as a compromise.” was added at the end of Section 3.
ROC curves in Figure 5 and 8 were an integral reflection of image filtering effect (the definition of ROC was added in Section 4). When drawing ROC curves, a threshold value was used. This threshold value swept from the minimum value of image gray to the maximum value, which was bound to include the threshold value used in further detection, but it was not necessarily related to . It was not to draw ROC curves through the sweep of .
Point 8: In section IV, five indices, SNR, SCR, BSF, ROC, and AUC are used to evaluate different methods. It would be beneficial to include the mathematical equations for how they are defined.
Response 8: Considering the Reviewer’s suggestion, we have added the definitions and mathematical equations of these indices. The following words were added in the Section 4.
“The definitions of these indexes are as follows:
(12)
where is the mean of the target gray level, is the mean of the background gray level, and is the standard deviation of the background gray level. SNR mainly reflects the correlation between the target gray level and the background gray level. The larger the SNR, the smaller the correlation between the target and the background, and the less the target is disturbed by the background.
(13)
where is the mean of the target gray level and is the mean of the background gray level. SCR mainly reflects the difference between the target gray level and the background gray level. The bigger the SCR, the bigger the gray difference between the target and the background.
(14)
where and are the standard deviations of background gray level before and after filtering, respectively. The larger the BSF, the stronger the suppression of the background after filtering.
For the filtered residual image, the ROC curve of the detection algorithm can be drawn by changing the detection threshold and traversing the probability of false alarm to obtain the corresponding probability of detection, taking as the horizontal axis and as the longitudinal axis. and are defined as:
(15)
where represents the number of pixels that the algorithm detects as the correct target, represents the number of real pixels of the target, represents the number of pixels where the algorithm detects the wrong target, and represents the number of real pixels of the background. Under the same , if the of the algorithm is higher than that of the others, it means the algorithm has better performance.
The area AUC under the ROC curve can be divided into several trapezoids. Let the point on the ROC curve be set as , where is the total number of points on the ROC curve. AUC can be expressed as:
(16)
The larger the AUC value, the better the performance of the algorithm and the better the detection performance.”
Point 9: At the end of the manuscript, the author contributions are not filled out.
Response 9: Considering the Reviewer’s suggestion, we have added the Author Contributions and the Conflicts of Interest at the end of the manuscript.
Point 10: I would recommend some more citations too:
On line 25, where the authors cite 2D reconstruction algorithms, I think it would be good to cite some of the recent microscopy reconstruction work:
Y. Rivenson, Y. Wu, H. Wang, Y. Zhang, A. Feizi, and A. Ozcan, “Sparsity-based multi-height phase recovery in holographic microscopy,” Sci. Rep. 6(1), 37862 (2016).
C. Fournier, F. Jolivet, L. Denis, N. Verrier, E. Thiebaut, C. Allier, and T. Fournel, “Pixel super-resolution in digital holography by regularized reconstruction,” Appl. Opt. 56(1), 69–77 (2017).
Zhen Xiong, Jeffrey E. Melzer, Jacob Garan, and Euan McLeod, “Optimized sensing of sparse and small targets using lens-free holographic microscopy,” Optics Express, 26 (20), 25676-25692 (2018).
On lines 73-79, references for these different matrices should be given. If they are all covered in one or two review articles, or a textbook, that would suffice.
On line 230, references for the IAA and SL0 algorithm should be given.
Response 10: We are extremely grateful to Reviewer for this recommend.
“[7]Y. Rivenson, Y. Wu, H. Wang, Y. Zhang, A. Feizi, and A. Ozcan, “Sparsity-based multi-height phase recovery in holographic microscopy,” Sci. Rep. 6(1), 37862 (2016).
[8]C. Fournier, F. Jolivet, L. Denis, N. Verrier, E. Thiebaut, C. Allier, and T. Fournel, “Pixel super-resolution in digital holography by regularized reconstruction,” Appl. Opt. 56(1), 69–77 (2017).
[9]Zhen Xiong, Jeffrey E. Melzer, Jacob Garan, and Euan McLeod, “Optimized sensing of sparse and small targets using lens-free holographic microscopy,” Optics Express, 26 (20), 25676-25692 (2018).” was added in References.
“[20] Jingwen Yang, Lei Liu and Xiaobo Qu. Compressive Sensing and Its Applications[M]. National Defence Industry Press, 2015.10:6-70.” was added in References for different measurement matrices.
“[4] M.J. Jahromi and M.H. Kahaei. Two-dimensional iterative adaptive approach for sparse matrix solution[J]. Electronics Letters. 2nd January 2014 Vol. 50 No. 1 pp. 45–47
[5] Aboozar Ghaffari, Massoud Babaie-Zadeh, Christian Jutten. Sparse Decomposition of Two Dimensional signals. ICASSP. 2009:3157-3460” was added in References for the IAA and SL0 algorithm.
Other changes:
1. The numbering style of section changed from “I, II,......” to “1,2,......”.
2. According to the other Reviewer’s comment, We have deleted “non-reconstruction” and replaced by “compressive domain”. The title of the article was revised to read “Two-Dimensional Adaptive Target Detection Algorithm in Compressive Domain”. And the name of our algorithm was revised to “two-dimensional compressive domain adaptive threshold (2D CDAT) algorithm”.
3. We have re-written the Section 2 according to the other Reviewer’s suggestion. In the Section 2, the review of 1D measurement model was deleted, and the describe of sparsity bases measurement matrixes was added at the end of Section 2.
We tried our best to improve the manuscript and made some changes in the manuscript. These changes will not influence the content and framework of the paper. And here we did not list the changes but marked in red in revised paper. We appreciate for Reviewer’ warm work earnestly, and hope that the correction will meet with approval.
Once again, thank you very much for your comments and suggestions.

Round 2
Reviewer 2 Report
I believe that the authors have addressed all of my points from the previous review and recommend publication.